# Cancer Associated Macrophage-like Cells Are Prognostic for Highly Aggressive Prostate Cancer in Both the Non-Metastatic and Metastatic Settings

**DOI:** 10.3390/cancers15143725

**Published:** 2023-07-22

**Authors:** Daniel J. Gironda, Raymond C. Bergan, R. Katherine Alpaugh, Daniel C. Danila, Tuan L. Chuang, Brenda Y. Hurtado, Thai Ho, Daniel L. Adams

**Affiliations:** 1Department of Cancer Biology, Wake Forest School of Medicine, Winston-Salem, NC 27157, USA; 2Division of Life Sciences, Rutgers, The State University of New Jersey, New Brunswick, NJ 08901, USA; 3Creatv MicroTech, Inc., Monmouth Junction, NJ 08852, USA; 4Department of Internal Medicine, University of Nebraska Medical Center, Omaha, NE 68198, USA; 5Fox Chase Cancer Center, Philadelphia, PA 19111, USA; 6Genitourinary Oncology Service, Memorial Sloan Kettering Cancer Center, New York, NY 10065, USA; 7Department of Medicine, Weill Cornell Medical College, New York, NY 10065, USA; 8Mayo Clinic Cancer Center, Phoenix, AZ 85054, USA

**Keywords:** prostate cancer, biomarker, prognostic, minimally invasive, circulating tumor cell, liquid biopsy

## Abstract

**Simple Summary:**

Prostate cancer (PCa) patient mortality rates remain high in patients with aggressive disease and the overtreatment of indolent tumors remains a major issue. Prostate-specific antigen (PSA), a standard PCa blood biomarker, is limited in its ability to differentiate disease subtypes, which results in the overtreatment of non-aggressive indolent disease. Recently cancer-associated macrophage-like cells (CAMLs), a cancer-specific polyploid circulating stromal cell, was found in the blood of patients with PCa. Further, it has been suggested that engorged CAMLs ≥ 50 μm in cytoplasmic diameter are associated with aggressive tumor subtypes and worsened patient outcomes, which may aid PSA for patient stratification. To expand upon previous research, we hypothesized that monitoring CAML size, in combination with PSA, may aid in differentiating indolent, non-aggressive, and highly aggressive PCas by adding biological information that complements traditional clinical biomarkers, thereby guiding treatment for PCa.

**Abstract:**

Despite advancements in the early-stage detection and expansion of treatments for prostate cancer (PCa), patient mortality rates remain high in patients with aggressive disease and the overtreatment of indolent disease remains a major issue. Prostate-specific antigen (PSA), a standard PCa blood biomarker, is limited in its ability to differentiate disease subtypes resulting in the overtreatment of non-aggressive indolent disease. Here we assess engorged cancer-associated macrophage-like cells (CAMLs), a ≥50 µm, cancer-specific, polynucleated circulating cell type found in the blood of patients with PCa as a potential companion biomarker to PSA for patient risk stratification. We found that rising PSA is positively correlated with increasing CAML size (r = 0.307, *p* = 0.004) and number of CAMLs in circulation (r = 0.399, *p* < 0.001). Over a 2-year period, the presence of a single engorged CAML was associated with 20.9 times increased likelihood of progression (*p* = 0.016) in non-metastatic PCa, and 2.4 times likelihood of progression (*p* = 0.031) with 5.4 times likelihood of death (*p* < 0.001) in metastatic PCa. These preliminary data suggest that CAML cell monitoring, in combination with PSA, may aid in differentiating non-aggressive from aggressive PCas by adding biological information that complements traditional clinical biomarkers, thereby helping guide treatment strategies.

## 1. Introduction

Prostate cancer (PCa) is the most common malignancy among men, and it is predicted that 1 in 6 men will develop prostate cancer during their lifetime [1,2]. It is recognized that there is a spectrum of PCa ranging from indolent to aggressive. Where PCa lies on this spectrum is determined by an integrated consideration of several parameters, most commonly PSA, microscopic appearance (i.e., grade group, as defined by Gleason pattern), and stage [3,4]. Despite advances in our understanding of the foundational biology of PCa (i.e., pathological risk factors, PSA testing, associated oncogenes, prognostic nomogram), differentiating aggressive and non-aggressive neoplasms prior to treatment initiation remains elusive as symptomology does not present until later stages of the disease, with some men never presenting symptoms until the development of widespread metastasis [3,4,5]. This situation has led to both overtreatment of ultimately indolent disease, with associated treatment-related side effects, and undertreatment of aggressive disease, with associated poor outcomes [3,4,5,6]. Biomarkers that can aid in differentiating aggressive from indolent PCa are needed, which may have a positive clinical impact on patient outcomes.

Prognostication for PCa with PSA has been a topic of controversy; despite improvements in patient survival as a result of PSA testing, the over-diagnosis of indolent disease is prevalent [7,8]. PSA collection is performed using blood plasma (i.e., a liquid biopsy), with PSA levels between 4 and 10 ng/mL being considered the “gray zone” for men in their sixties and older, with <4 ng/mL considered negative, and ≥10 ng/mL necessitating PCa surveillance [9,10,11]. However, PSA measurements are age-adjusted and PSA ≥ 2.5 ng/mL is considered abnormal in men between their forties and fifties [12]. Typically, increasing PSA levels at diagnosis are suggestive of actively growing, aggressive disease [13]. Fluctuations in PSA are measured by PSA kinetic measurements such as velocity (ng PSA/mL/year) and doubling time (the number of months for PSA to increase two-fold). However, utilizing PSA kinetics as a determinant of aggressive disease is limited, as there is no universal PSA ratio of harm versus benefit that can reliably differentiate aggressive and indolent PCa [14,15]. The combination of PSA with a prognostic nomogram has shown to increase the prognostic accuracy in identifying indolent PCa over PSA alone. However, the addition of a prognostic nomogram has some limited utility, being that a surgical procedure is required for analysis and the nomogram model is not highly sensitive in prognosticating indolent disease [16]. To mitigate this issue, the 2021.1 updated NCCN guidelines suggest PCa classification be based on tumor burden (i.e., T-score, ECOG, Gleason score) over PSA, as patients with aggressive disease can present PSA < 10 ng/mL [17,18]. Due to these reasons, PSA testing is no longer recommended by the U.S. Preventative Services Task Force for men over 70 years old, as the harm of treatment outweighs the benefit of therapy [19]. Given that there is no consistent threshold for PSA in predicting tumor aggressiveness, companion assays that can better identify refractory tumors are of high demand at the diagnostic stage [5]. 

The integration of liquid biopsies has become a common tool in cancer prognostication as they are non-invasive, can monitor tumor responses during treatment, and are cost-effective [20]. Beyond PSA, non-invasive blood-based biomarkers such as circulating micro-RNA (miRNA), cell-free DNA (cfDNA), and circulating tumor cells (CTCs) have been identified as having prognostic value in prostate, breast, colorectal, and non-small- cell lung cancers [21,22]. Although promising, sequencing of miRNA and cfDNA has been limited, as determining tumor mutational burden depth from cfDNA and miRNA is limited by the amount of input material, in addition to large-scale sequencing studies being cost-prohibitive [23,24]. CTCs are an extensively studied biomarker associated with a terminal pathological discourse in PCa but they have shown little clinical utility in the localized disease setting due to low cell presence (~0–20%) [22,25,26,27]. Molecular markers within CTCs, such as expression of the androgen-receptor splice-variant 7 (AR-V7), have been acknowledged as predictive biomarkers among chemo-resistant phenotypes in castration-resistant PCa [28]. More specifically, AR-V7 is a clinically predictive biomarker associated with poor therapeutic benefit to anti-androgen therapies, that is, enzalutamide or abiraterone [28,29]. Leveraging liquid biopsies for the molecular quantification of AR-V7 and PSA has been evaluated and proven, yet low CTC frequency in the localized setting remains a limiting factor in how this technology can be applied to clinical populations [29,30,31,32]. Given the rarity of CTCs in low-stage PCa patients, more common blood-based biomarkers are needed for identifying molecular vulnerabilities in both the metastatic and non-metastatic settings for drug-target selection and treatment optimization.

Cancer-associated macrophage-like cells have been identified as a cancer-specific, myeloid-derived circulating cell that appears to be an independent prognostic indicator of tumor aggressiveness across multiple solid-tumor malignancies [33,34,35,36,37]. Uniquely, CAMLs are solely identified in human cancer patients’ blood and appear absent in healthy controls [25,35,36,37,38,39,40]. Cytological identification of CAML cells is dependent on morphological features such as cell cytoplasmic diameter ≥ 30 µm, the presence of polyploid nuclei, as well as the phenotypic expression of CD45, Cytokeratins 8, 18, 19 and DAPI [35]. Independent of cancer etiology, engorged CAMLs (≥50 µm) have shown clinical utility as a prognostic and predictive biomarker for worse outcomes in breast, non-small-cell lung, esophageal, and pancreatic cancers [36,37,40,41,42]. Further, the presence of CAMLs and their size association to clinical outcomes have been investigated and independently validated by a number of groups [43,44,45,46,47]. While ≥50 µm CAMLs appear to predict decreased survival across multiple solid-tumor types, there is an unexplained phenomenon in which hyper-engorged CAMLs (≥100 μm) appear to predict multi-organ-site metastatic spread and even poorer survival [40,48]. To expand upon established research across a variety of solid tumors, and as an avenue to complement PSA, we evaluated engorged CAMLs’ ability to predict worse patient outcomes in both localized and metastatic PCa, prior to the initiation of new treatment for current disease to determine the CAML cell’s clinical utility across all stages of PCa.

## 2. Materials and Methods

### 2.1. Study Design

Anonymized peripheral blood samples were collected with local Institutional Review Board (IRB) approvals from Oregon Health and Science University (IRB00011862), Northwestern University (STU0019487), Memorial Sloan Kettering Cancer Center (90-040A), Mayo Clinic Cancer Center (08-000980), and Fox Chase Cancer Center (99-802 and 11-866) with patients’ written informed consent prior to the initiation of this study. Patient enrollment information for each respective institution is provided in Appendix A. Once approved, we initiated a 2-year multi-center prospective single-blind pilot study composed of n = 92 men with non-metastatic PCa (n = 50) or metastatic PCa (n = 42) to examine the clinical utility of CAMLs (≥50 µm) as they relate to rates of progression-free survival (PFS) and overall survival (OS). Pathological diagnosis of PCa was defined by a board-certified physician following the American Joint Committee on Cancer (AJCC) 7th edition guidelines. Broadly defined, Stage I PCa includes primary tumors that have not invaded beyond the prostate, are well-differentiated, and present negligible to low PSA levels. Stage II PCa is also confined to the prostate with moderate to poor differentiation and medium PSA levels. Stage III PCas have invaded beyond the prostate into nearby tissues (i.e., bladder, rectum), are poorly differentiated, and express high levels of PSA. Stage IV PCa has spread beyond the prostate to local lymph nodes, regional lymph nodes, bones, or other parts of the body, are poorly differentiated, and are rapidly growing.

Patient samples were collected from January 2012 through to October 2016. BL samples were categorized as any blood draw taken within 1–4 weeks after pathological confirmation of newly diagnosed PCa, or prior to the initiation of a new treatment for current disease. If possible, patients volunteered for follow-up time point blood draws, T1 and T2, which were taken ~2 weeks (T1) into treatment, or at the first blood draw after the completion of treatment (T2), respectively. Randomized and anonymized patient blood samples (7.5 mL) were collected in CellSave preservative vacutainer tubes (Menarini Silicon Biosystems, Huntington Valley, PA, USA) and prepared according to standard operating procedures at each respective institution (see CAML Isolation and Enumeration below). Once collected, peripheral whole blood specimens were shipped to Creatv Microtech Inc.’s (Monmouth Junction, NJ, USA) clinical laboratory for circulating cell enumeration and analysis. Due to limited availability of personnel during the COVID-19 pandemic, there was an ~2-year gap between the last blood procurement and the analysis of these findings.

### 2.2. CAML Isolation and Enumeration

PCa patient blood samples were collected into 7.5 mLCellSave preservative tubes (Menarini Silicon Biosystems) at each respective institution, maintained at room temperature, then shipped overnight and processed within 96 h after collection at Creatv Microtech’s core laboratory. Creatv Microtech Inc. did not perform any blood collection, just blood sample processing and downstream analysis. Preserved samples were filtered via a CellSieve^TM^ Microfiltration assay on a low-flow vacuum system. The mechanism of CellSieve^TM^ microfiltration works by size exclusion of whole-blood components ≥7 μm. Post filtration, microfilters were post-fixed, permeabilized, and then stained with an antibody cocktail of Cytokeratins 8, 18, and 19 tagged with FITC, and CD45+ tagged with Cy5. Post-staining, filters were washed and then mounted with Fluoromount-G with DAPI (Southern Biotech, Birmingham, AL, USA). CTCs were identified using the standard CTC definition—by expression of Cytokeratins 8, 18, and 19, and CD45-, as previously described by Adams et al. [49]. Once stained, all samples were stored at 4 °C and quantified within 3 months post processing. Identification and enumeration of CAMLs was performed by a trained cytologist at a single microscopy lab on an Olympus BX54WI Fluorescent Microscope with Carl Zeiss AxioCam and Zen 2011 Blue (Carl Zeiss, White Plains, NY, USA). CAMLs were identified as 30–300 μm in size, DAPI- positive polyploid nuclei, surface markers of CD45, and a diffuse expression of cytokeratin, with negative isotypes previously described [33,34,35,36,37,39,41,49,50]. Apoptotic and denucleated cells were not included in our analyses.

### 2.3. Statistical Analyses

After confirmation of clinical data unblinding from each clinical site, data analyses were performed independently using MATLAB 2020a and Prism 8.0.2. The primary endpoints of this study were to determine CAMLs’ ability to prognosticate for worsened PFS and OS, and to evaluate CAML size’s relationship to disease stage. Survival functions were evaluated from the time of BL draw until the time of either progression, death, or censored at last known date of contact. Patients with evidence of radiographic metastasis at the time of diagnosis and then later confirmed with metastatic disease after the induction of new treatment were grouped and analyzed within the metastatic patient cohort. Cox Proportional-Hazard univariate analyses were conducted for all known clinical variables among the entire 92 patient cohort to determine statistically significant predictors of patient outcomes across all stages of disease. Once all predictive clinical variables were identified across non-metastatic and metastatic patients, multivariate regression analysis was conducted to determine the most statistically significant independent predictors for both PFS and OS across all patients (Appendix A). Kaplan–Meier survival estimates were then undertaken to determine CAMLs’ ability to predict worsened PFS and OS. Single-factor ANOVA was used to compare circulating cell frequency between metastatic and non-metastatic patients, as well as the variance between individual pathological stages. PSA sensitivity at BL was examined at ≥4 ng/mL and ≥10 ng/mL, and prediction of worsened PFS and OS used PSA cutoffs of ≥10 ng/mL, ≥20 ng/mL, and ≥50 ng/mL [2,51,52,53]. Patients with no CAMLs present or counts of 0 were included in all statistical analyses when comparing groups. Statistical significance was defined to be any *p* value ≤ 0.05 and statistical trending was any *p* value ≤ 0.15 and >0.05.

## 3. Results

### 3.1. Patient Demographics

From January 2012 to October 2016, we enrolled n = 92 PCa patients with newly diagnosed disease, biochemically recurrent, or progressing disease. Out of (n = 92) the total patients, 15% (n = 14/92) were stage I, 30% (n = 28/92) stage II, 9% (n = 8/92) stage III, and 46% (n = 42/92) stage IV (Table 1). Prostate adenocarcinoma comprised 92% (n = 46/50) of the non-metastatic cohort and the histology was undetermined in 8% (n = 4/50) of non-metastatic patients. In the non-metastatic group, 72% (n = 36/50) were newly diagnosed untreated and 16% (n = 8/50) were defined as biochemically recurrent prior to initiation of second-line therapy, 14% (n = 7/50) had received prior chemotherapy, and 28% (n = 14/50) had undergone androgen-deprivation therapy (ADT). In metastatic patients, 7% (n = 3/40) were newly diagnosed untreated with 2% (n = 1/40) defined as progressive PCa by biochemical recurrence, 71% (n = 30/42) had received prior chemotherapy, and 93% (n = 39/42) had received ADT. Prostate adenocarcinoma was present in 93% (n = 39/42) of metastatic patients, 2% (n = 1/42) had neuroendocrine PCa, and 5% (n = 2/42) were of unknown histology.

### 3.2. CAML Cell Presence versus Conventional PCa Bioassays

Prior to the induction of treatment for new disease, or progressive disease starting a new line of therapy, CAMLs were identified in 79% (n = 71/90) of available BL blood samples (average: ~5 CAMLs/7.5 mL). Two samples failed due to blood clotting during microfiltration. CAML presence had a sensitivity of 78% (n = 39/50) in the non-metastatic cohort, and 80% (n = 32/40) in the metastatic population, with no statistical difference between groups (*p* = 0.820) (Figure 1). Clinically, this finding could be of value as CAML cells are highly sensitive and can be identified ubiquitously across PCa patients independent of non-metastatic or metastatic status. Additionally, non-metastatic PCa averaged 3 CAMLs/7.5 mL, whereas mPCa averaged 6 CAMLs/7.5 mL (*p* = 0.108). Though statistically non-significant, the average number of CAMLs in circulation appears to be twice as high in mPCa over patients with non-metastatic. The frequency of all circulating tumor cells examined in this study (i.e., CTCs, EMTs, CAMLs) is provided in Appendix A.

Stratifying individual pathological stage, CAML presence was 57% (n = 8/14) stage I, 82% (n = 23/28) stage II, 100% (n = 8/8) stage III, and 80% (n = 32/40) of mPCa. CAML sensitivity between pathological stages found that CAMLs are less common in stage I disease than stages II and IV (*p* = 0.086 and *p* = 0.073, respectively), and are significantly less frequent compared to stage III (*p* = 0.030) (Figure 1). Average CAML number in stages I, II, III, and IV patients contained 3, 3, 6, and 6 CAMLs/7.5 mL blood, respectively, with no statistical difference between groups. Pearson correlation analysis was then undertaken to determine if there is any relationship between increasing pathological stage and the number of CAML cells in circulation. We found that there was a statistically trending, weak positive association between advancing pathological stage and the increasing number of CAML cells in circulation (r = 0.181, *p* = 0.087). Based on these preliminary data, CAMLs appear more common in PCa with progressing or advanced disease, and the number of CAMLs in circulation does not differentiate individual pathological stages but may differentiate local and advanced PCa.

CAMLs were found to be more sensitive over CTCs in circulation (79% CAMLs vs. 21% CTCs, *p* < 0.001) across all stages of disease. Comparing CAML and CTC presence among individual stages, CAMLs are more common than CTCs in stage I disease (57% vs. 7%, *p* = 0.003), stage II (82% vs. 14%, *p* < 0.001), stage III (100% vs. 38%, *p* = 0.004), and stage IV (80% vs. 28%, *p* < 0.001). When comparing non-metastatic and metastatic cohorts, CAMLs were statistically more sensitive than CTCs in both non-metastatic (78% vs. 16%, *p* < 0.001) and metastatic (80% vs. 28%, *p* < 0.001) cohorts. Overall, CAMLs were found to be the more sensitive circulating cell type across all stages of PCa over CTCs (Figure 1).

We then compared CAML sensitivity to PSA thresholds ≥4 ng/mL or ≥10 ng/mL to see if they can be used to supplement the PSA “gray zone” for identifying aggressive PCa. Among all patients with BL PSA counts (n = 89), 79% (n = 70/89) had PSA levels ≥ 4 ng/mL and 49% (n = 44/89) ≥10 ng/mL. Single-factor ANOVA found that CAML cells were significantly more sensitive in PCa when PSA was ≥10 ng/mL (79% vs. 49%, *p* < 0.001) but were similar to PSA values ≥ 4 ng/mL (79% vs. 79%, *p* > 0.50). This pattern held in non-metastatic disease, as CAMLs appeared more sensitive than PSA at ≥10 ng/mL (78% vs. 35%, *p* < 0.001) and similar to PSA at ≥4 ng/mL (78% vs. 76%, *p* > 0.50). In metastatic disease, there was no statistical difference between CAMLs and PSA values. When examining individual stages for CAML sensitivity and PSA values ≥ 4 ng/mL, CAMLs’ sensitivity was higher in stage III disease (100% vs. 75%, *p* = 0.150) but the assays were equally sensitive in other stages. Further, CAMLs were more sensitive than PSA ≥ 10 ng/mL in stage II (82% vs. 37%, *p* < 0.001) and stage III (100% vs. 38%, *p* = 0.004) patients but not the more sensitive assay in stage I disease (57% vs. 29%, *p* = 0.136) nor stage IV (80% vs. 83% *p* = 0.209). Given that CAMLs are cancer-specific and are found ubiquitously in patient blood, with similar sensitivity to low PSA ≥ 4 ng/mL levels, these data suggest that implementation of CAML isolation in tandem with PSA quantification may add diagnostic sensitivity versus PSA alone, though follow-up studies will be needed to better elucidate this relationship.

### 3.3. CAML Size Differentiates Local and Advanced Disease

Increasing CAML size (beyond 50 μm) has been implicated in solid-tumor pathogenesis and decreased patient survival across multiple solid cancers [36,39,40,41,48]. To examine this pattern in PCa, we compared all BL samples, finding the average max CAML size was statistically larger in patients with metastatic disease over localized PCa (78 μm metastatic vs. 35 μm localized, *p* < 0.001) (Figure 2). We then compared CAML size for each stage, finding that average CAML size increased with advancing disease, stage I averaging 23 μm, stage II 33 μm, stage III 65 μm, and stage IV 78 μm. Single-factor ANOVA comparing sizes based on pathological stage found no statistical difference in max CAML size between stage I and II patients, as well as no difference between stage III and IV patients. However, CAML cells in patients with stage III or IV disease were statistically larger than in stages I or II by single-factor ANOVA (average = 75 μm vs. 30 μm, *p* < 0.001). Pearson correlation analysis was then undertaken to determine the relationship between increasing CAML size and increasing pathological stage, which found that there was a statistically significant, positive correlation between increasing CAML size and advancing pathological stage (r = 0.415, *p* < 0.001). We then divided CAML sizes into three subgroups to better elucidate the relationship with pathological stage at BL: (1) 0 CAMLs or CAMLs < 50 μm, (2) CAMLs ≥ 50–99 μm, and (3) CAMLs ≥ 100 μm (Figure 2). We found that stage I and II patients appeared to have a nearly identical distribution of CAML sizes, with most patients having 0 CAMLs present or <50 μm CAMLs. In contrast, larger CAMLs (i.e., ≥50 μm and ≥100 μm) were more commonly found in circulation among stage III and IV patients. We then conducted Pearson correlation to identify a relationship between the number of CAML cells in circulation and increasing CAML size, and found that there is a statistically significant, moderate positive correlation between the number of CAML cells in circulation and the size of the largest CAML cell (r = 0.653, *p* < 0.001). Interestingly, CTC presence, a phenomenon found in advanced disease, appeared to have a relationship to engorged CAML presence in both the non-metastatic and metastatic settings (Appendix A). This suggests that engorged CAMLs are more sensitive among advanced PCas and may coincide with CTC intravasation.

### 3.4. Engorged CAMLs Found Prior to Treatment Predict for Early Mortality

At BL sampling, 41% (n = 37/90) of patients presented with ≥50 μm CAMLs vs. those with <50 μm CAMLs, which predicted for shorter median progression-free survival (mPFS = 7.9 vs. >24 months) as well as shorter median overall survival (mOS = 17.4 vs. >24 months). Cox-Fit Proportional analysis found that engorged ≥50 μm CAMLs at BL were able to prognosticate for worsened PFS (HR = 7.5, 95%C.I. = 3.7–15.3, *p* < 0.001) and worsened OS (HR = 13.3, 95%C.I. = 5.5–32.5, *p* < 0.001) (Appendix A).

After examining CAMLs’ relationship to clinical outcomes, we then analyzed the non-metastatic PCa and metastatic PCa cohorts separately. In non-metastatic, 23.5% (n = 12/50) of patients had ≥50 μm CAMLs. mPFS and mOS could not be calculated, as too few patients had clinical events within the 2-year endpoint, PFS (12%, n = 6/50) and OS (6%, n = 3/50). Although the mPFS and mOS could not be calculated, it was found that ≥50 μm CAML presence in non-metastatic disease did significantly predict for worsened PFS (HR = 20.9, 95%C.I. = 2.7–159.7, *p* = 0.016) but not OS (HR = 9.7, 95%C.I. = 0.7–135.1, *p* = 0.306) (Figure 3). The lack of significance in OS was likely a result of too few patients dying within the study time frame, and larger longer-term studies may be required to provide a significant endpoint based on CAML engorgement.

In the metastatic cohort, patients with mPCa were found to have ≥50 μm CAMLs in 63% (n = 25/40) of samples, which predicted for shorter mPFS (4.7 vs. 12.8 months) as well as shorter mOS (16.2 vs. >24 months). Comparative analyses found that engorged CAMLs significantly predicted for expedited patient progression (HR = 2.4, 95%C.I. = 1.2–4.9, *p* = 0.031) as well as expedited death (HR = 5.4, 95%C.I. = 2.2–13.4, *p* < 0.001) in mPCa (Figure 3). These results demonstrate that engorged CAMLs in circulation may be predictors of worse survival across all stages of PCa, with non-metastatic patients having statistically higher rates of progression and metastatic patients having rapid progression and death.

To explore previous literature [36,39,40,41,48] on hyper-engorged CAMLS (≥100 μm) for predicting severely worse outcomes in patients, we investigated how these hyper-engorged CAMLs relate to PCa patient clinical response. First, we examined ≥100 μm CAML presence among all patients and found that 14% (n = 13/90) of BL samples had the hyper-engorged CAML phenotype. Survival analysis of ≥100 μm CAMLs were found to predict for shorter mPFS (4 vs. >24 months) and shorter mOS (8.7 vs. >24 months) when compared to patients with CAMLs <100 μm in diameter (Figure 4). Further analysis then confirmed that ≥100 μm CAMLs predicted for worsened PFS (HR = 12.1, 95%C.I. = 4.1–40.0, *p* < 0.001) and worse OS (HR = 71.1, 95%C.I. = 18.0–280.3, *p* < 0.001) (Figure 4). We then examined non-metastatic patients for evidence of hyper-engorged CAMLs, which were seen in 2% (n = 1/50) of all samples, preventing survival analysis from being run. Interestingly, this individual was diagnosed with stage IIIb biochemically recurrent adenocarcinoma who progressed and died within 16 months after identification of the ≥100 μm CAML. In the mPCa population, we found ≥100 μm CAMLs in 30% (n = 12/40) of BL samples which predicted for shorter mPFS (3.9 vs. 9.1 months) and shorter mOS (7.5 vs. 19.9 months). In evaluating ≥100 μm CAMLs for survival outcomes, it was found that ≥100 μm CAMLs were statistically significant predictors for worse OS (HR = 3.7, 95%C.I. = 1.3–10.1, *p* = 0.025) but not worse for PFS in metastatic PCa (Figure 4). It appears that hyper-engorged CAMLs may possibly predict for worse patient survival in metastatic PCa. Although hyper-engorged CAMLs were found in only 2% (n = 1/50) of non-metastatic PCa patients, anecdotally, the one patient died within two years.

### 3.5. CAML Size Tracking throughout Treatment

Following new treatment induction, n = 36 patients volunteered a midpoint treatment (T1) sample and n = 11 volunteered samples at treatment completion (T2). Engorged CAMLs were present in 28% (n = 10/36) at T1, which predicted for shorter mPFS (5.1 months vs. >24 months) and mOS (16.5 months vs. >24 months), as well as shorter overall PFS (HR = 12.2, 95%C.I. = 3.4–43.0, *p* < 0.001) and shorter OS (HR = 17.8, 95%C.I. = 3.9–80.5, *p* < 0.001). Due to insufficient samples available at T1 for non-metastatic patients (n = 17), survival analysis was not possible. However, metastatic patients with engorged CAMLs at T1 (47%, n = 9/19) had shorter mPFS (4.6 vs. 11.7 months) and shorter mOS (15.6 vs. >24 months), which statistically predicted for worsened PFS (HR = 4.1, 95%C.I. = 1.3–12.9, *p* = 0.030) and OS (HR = 6.3, 95%C.I. = 1.6–24.8, *p* = 0.024) at T1 (Appendix A).

Despite a sparse number of patients at the T2 timepoint, we conducted preliminary analyses (Appendix A). Among these patients, 27% (n = 3/11) had CAMLs ≥ 50 μm at T2 which associated with shorter mPFS (5.1 vs. 7.9 months) and shorter OS (8.6 vs. 20.8 months). Further, these patients trended towards worse OS (HR = 8.8, 95%C.I. = 1.1–70.9, *p* = 0.133) but did not predict for worse PFS (HR = 3.4, 95%C.I. = 0.6–24.8, *p* = 0.342). While this patient cohort was too small for proper clinical outcome analysis, these initial findings appear promising. Follow-up studies with larger patient populations, controlled treatment groups, and multiple timepoints throughout treatment will be necessary.

### 3.6. Multivariate Analysis

A multivariate analysis was used to compare all known significant variables for patient PFS and OS. Parameters for multivariate analysis were defined to be age ≥70 years, pT ≥ T2 (locally confined to the prostate), pN ≥ N1 (metastasis in a single regional lymph node < 2 cm), pM ≥ M1 (distant metastasis), Gleason score ≥ 8, PSA ≥ 50 ng/mL, ≥1 CTCs present, ≥3 CAMLs present, and CAMLs ≥ 50 μm (Appendix A). Among all patients, CAMLs ≥ 50 μm at BL were the most statistically significant independent predictor of PFS (*p* = 0.002) with PSA ≥50 ng/mL (*p* = 0.007), nodal spread (*p* = 0.011), metastatic status (*p* = 0.015), and Gleason score ≥8 (*p* = 0.048) also being significant independent indicators. In addition, engorged CAMLs (≥50 μm) were the most statistically significant independent predictor for worse OS (*p* = 0.006), followed by nodal spread (*p* = 0.028) and age ≥ 70 years (*p* = 0.041). Due to limited patients reaching the time to event, as well as limited population sizes, a multivariate analysis could not be run separately for the non-metastatic and metastatic cohorts. To better elucidate the prognostic significance of engorged CAMLs, larger validation studies in refined cohorts of PCa is warranted.

### 3.7. Analysis of PSA for Predicting PFS and OS

To best identify PSA thresholds for prognosticating PCa, we evaluated PSA levels ≥10 ng/mL, ≥20 ng/mL, and ≥50 ng/mL. Among all patients, 97% (n = 89/92) had available PSA counts at BL, with three patients lost during unblinding. Pre-treatment PSA levels among all patients found 17% (n = 15/89) had 10–20 ng/mL, 10% (n = 9/89) had 20–50 ng/mL, and 22% (n = 20/89) had ≥50 ng/mL. The combination of both non-metastatic and metastatic diseases found that all three PSA cutoffs were statistically significant predictors for worsened PFS and OS, with increasing PSA predicting shorter survival (Figure 4).

Among non-metastatic patients with available PSA counts (n = 49), 22% (n = 11/49) had PSA between 10 and 20 ng/mL, 4% (n = 2/49) between 20 and 50 ng/mL, and 8% (n = 4/49) >50 ng/mL. We found that all three PSA cutoffs were not significant predictors for worsened survival and all groups had mPFS and mOS > 24 months.

Metastatic patients had (n = 40) available BL PSA counts, with 10% (n = 4/40) presenting PSA levels between 10 and 20 ng/mL, 18% (n = 7/40) between 20 and 50 ng/mL, and 40% (n = 16/40) having ≥50 ng/mL. It was found that PSA levels ≥10 ng/mL had shorter mPFS (6.3 vs. 14.7 months) and mOS (13.7 vs. 23.2 months) but were not statistically significant predictors for PFS or OS (Figure 4). Similarly, patients with PSA ≥20 ng/mL had shorter mPFS (7.3 vs. 11.5 months) and mOS (13.7 vs. 23.2 months) but did not reach statistical significance. However, patients with PSA ≥50 ng/mL had shorter mPFS (4.2 vs. 13.6 months) but statistically trended toward worsened PFS (HR = 2.5, 95%C.I. = 1.1–5.7, *p* = 0.051). Further, PSA ≥50 ng/mL appeared to have shorter mOS (10.1 vs. >24 months) and was a statistically significant predictor for worsened OS (HR = 3.9, 95%C.I. = 1.4–10.5, *p* = 0.015) in the metastatic setting (Figure 4). Follow-up monitoring of increasing PSA at T1 and T2 found no statistical significance in predicting worsened survival.

To determine the relationship between PSA and CAML cells in circulation for prognosticating patient survival, we looked at different thresholds of BL PSA in combination with CAMLs < 50 μm or CAMLs ≥ 50 μm to predict patient outcomes (Appendix A). Engorged CAMLs were found to predict worse PFS and OS in patients with patients with PSA < 4 ng/mL (Appendix A), PSA < 10 ng/mL (Appendix A), PSA < 20 ng/mL (Appendix A), and PSA < 50 ng/mL (Appendix A). However, CAMLs were unable to stratify PFS and OS in patients with PSA ≥ 50 ng/mL (Appendix A). We then sought to determine if there is any correlation between rising PSA and increasing CAML number or rising PSA with increasing CAML size. Pearson correlation analysis identified a statistically significant, weak positive correlation between increasing PSA concentration and CAMLs in circulation, as well as a statistically significant, weak positive correlation between rising PSA and increasing CAML size (Appendix A). To examine the opposite relationship, we then sought to determine if adding PSA to CAML sizes < 50 μm or ≥50 μm can better stratify patient responses (Appendix A). In patients with CAMLs < 50 μm, PSA values ≥ 50 ng/mL added prognostic value for PFS (HR = 97.8, n = 6 vs. 44), whereas PSA levels ≥ 20 ng/mL (HR = 464.9, n = 11 vs. 39) and ≥50 ng/mL (HR = 7.1 × 10^5^, n = 6 vs. 44) added prognostic value for OS. Interestingly, in patients with engorged CAMLs, only PSA levels ≥ 50 ng/mL added prognostic value for OS (HR = 3.1, n = 13 vs. 24).

We then examined 10 mPCa patients with CAML and PSA tracking data for BL, T1, and T2 to compare the clinical accuracy of ≥50 μm CAMLs and rising PSA in prognosticating 24-month OS. Among these 10 patients, CAMLs matched PSA tracking in 40% of samples (i.e., patient CAMLs remained <50 μm and PSA was stable/declining, or the patient CAMLs were ≥50 μm and PSA increased) and this was associated with the likelihood of death within 24 months. In 50% of these patients, CAMLs were more accurate at predicting survival than PSA tracking over 24 months (i.e., patients had ≥50 μm CAMLs and PSA declined, or patients had <50 μm CAMLs and PSA increased). These data strengthen the argument that there is a correlation between PSA levels and CAML cells in circulation, and this correlation may better stratify patient survival than PSA alone.

## 4. Conclusions

Clinically translatable assays that can differentiate aggressive, non-aggressive, and indolent PCa while actively monitoring therapeutic response remain in demand. As non-invasive, blood-based biomarkers that are present among multiple solid-tumor malignancies, CAMLs may have clinical value as a prognostic that identifies pre-treated patients with more aggressive tumor types. Here, we ran a multi-institutional pilot study to examine what, if any, clinical utility CAMLs may have in local and advanced PCa. We found that CAMLs are a highly sensitive biomarker which appears frequently (79%) among PCa patients. Further, although not statistically significant, CAML sensitivity is high across all pathological stages of PCa when compared with traditional PCa biomarkers. However, a potential advantage of CAML screening over the latter is that there may be a relationship between the frequency of larger CAMLs in circulation and increasingly aggressive PCa as measured by pathological staging. Survival analysis based off of CAML size suggests that as CAML engorgement increases ≥50 μm, PCa patients present a dismal clinical discourse irregardless of non-metastatic or metastatic presentation. Further, patients with CAML cells ≥100 μm were found to have even worse survival, remarkably so in the metastatic setting. Multivariate analysis of our data found that engorged CAMLs (≥50 μm) at BL were the most statistically significant independent predictors of worse PFS and OS across all 92 patients, independent of local or advanced disease status (Appendix A). The preliminary results of this prospective pilot study suggest that engorged CAMLs predict for aggressive pathological maturation as measured by faster patient progression and death, even before the induction of new treatment for PCa.

A common issue with PSA screening is over-staging indolent disease [7] and high prevalence of false positives (upwards 70%) [54], causing the overtreatment of non-aggressive neoplasms. This results in increased adverse biopsy events (i.e., hematuria, pain, septicemia) and worse toxicities (i.e., vomiting, diarrhea, anemia) to standard of care treatment [5,7,21,55,56]. These issues have left a need for a companion biomarker to PSA to help in screening early PCa and for stratifying patients with more aggressive diseases. The high presence of CAMLs prior to resection and first-line treatment suggests that CAMLs might be useful in conjunction with PSA and could increase specificity in the diagnostic setting. While these initial findings are promising in the context of discriminating indolent versus aggressive PCa, additional studies which include other benign prostate conditions, such as BPH or low Gleason-scored prostate cancer (i.e., <6) should be evaluated. Though this hypothesis requires a number of prospective validation studies with healthy and non-malignant disease controls, CAML screening may provide additive diagnostic and prognostic information not captured by conventional PCa tests such as PSA doubling time and the PCa nomogram.

While CTCs are a heavily researched biomarker associated with poor PCa prognoses, CTCs in circulation are rare in localized disease and found in only ~25% of mPCa patients [16,22,26]. Here, we compared CTCs to CAMLs in circulation for clinical sensitivity, clinical relevance, and biological relationship. Our results suggest that CAMLs appear more commonly than CTCs across all pathological stages, and engorged CAMLs are a more accurate predictor for worsened outcomes. In line with prior literature, our CTC results did indicate worsened survival and were found with similar sensitivities to prior literature in non-metastatic disease (7.5–18.6%) [27,57] and mPCa (25–57%) [22]. Interestingly, the presence of CTCs seemed to relate to engorged CAMLs, with ≥50 μm CAMLs appearing in higher proportions in patients that had CTCs. This relationship between CAMLs and CTCs supports Adams’ PNAS findings in breast cancer, which described that CAMLs might initiate tumor cell transversion through endothelial junctions into circulation, and here, we identified a similar relationship in mPCa [35]. However, to evaluate this hypothesis, follow-up animal model studies must investigate the relationship between CTCs and CAMLs.

Although there is extensive literature in the CTC field, preclinical models do not yet exist in CAML research. CAML cell extraction, isolation, and enrichment have been found to be complicated. No study to date has demonstrated that CAMLs can be cultured ex vivo, and there is no confirmation to whether these cells undergo mitosis and can be immortalized. As such, the CAML cell type is only known to be human solid-tumor-specific, with no investigation done in non-human primates or other animal models, suggesting an undeveloped area of study. A major constraint of animal models in CAML research is the physical size of the CAML cell. For example, a 250 μm circulating cell would emulate an embolism in most rodent models’ vasculature suggesting larger animal models may be a more suitable avenue for testing.

Currently, there are two hypotheses as to the role of CAMLs in cancer pathogenesis —(1) CAMLs are disseminators of other circulating cancer cells via phagocytic attack on the extracellular matrix (ECM) of the primary tumor, helping promote the intravasation process of other tumor-initiating cells or (2) CAMLs are sloughed off of tumors themselves, and the high presence of large CAMLs in circulation is the result of widespread cancer cell dissemination from multiple anatomically distinct tumor sites. Given the phagocytic capabilities of the CAML cell, genetic material from the primary tumor [58] and subsequential distant metastatic sites may be resolved within CAMLs by next-generation sequencing techniques (i.e., single-cell transcriptomics, whole exome sequencing, whole genome bisulfite sequencing) to elucidate the clonal composition of multiple tumor sites without the need for surgical intervention. If this is the case, the combination of single-cell sequencing and whole exome sequencing on CAML cells may provide biological insight regarding differentially expressed gene signatures and mutations associated with chemo-resistance, chemo-sensitivity, and organ-specific tropism for locating distant metastases missed by conventional CT scans. Further, by leveraging whole genome bisulfite sequencing, methylated regions of the genome that encode for transcription factors may be identified to home in on what oncogenic signaling and cancer cell stemness pathways directly contribute to metastatic seeding.

Despite our finite understanding of the molecular origins of CAMLs, it appears that the phenomenon of CAML engorgement to ≥50 μm is related to aggressive local and advanced PCas which predicts worse patient outcomes. To date, the closest equivalent to CAML engorgement is the presence of CTC clusters promoting higher rates of metastasis in breast cancer. Specifically, transcriptomic analysis of CTC clusters compared to individual CTCs found that CTC clusters have a higher likelihood of DNA hypomethylation for stemness transcription factor binding sites such as OCT4, SOX2, NANOG, and SIN3A [59]. If this concept of increased cell volume promoting transcription factor activation is conserved in CAMLs, engorged CAMLs may have higher rates of DNA hypomethylation compared to small CAMLs which may promote EMT signaling cascades for metastatic spread, or tumor self-seeding in the context of recurrent disease [58]. Further, prospective studies on actionable targets, such as cell surface receptors or oncogenes (i.e., PD1 or AR-V7) in CAMLs may be an appealing avenue for shaping immunotherapy treatments if these results are validated. As stated above, AR-V7 expression is quantifiable on CTCs [28]. Given that CAMLs are a more common circulating cell type over CTCs, it would be of interest to identify AR-V7 alterations on CAMLs for aiding treatment selection in prospective clinical trials. With personalized medicine becoming the goal of oncology research, exploratory analysis of cell surface markers, signaling proteins, and transcription factors on CAMLs could be a valuable addition to the arsenal of prognosticating PCa pathogenesis. The results of this study suggest a possible relationship between CAML presence and patient clinical outcomes across all subtypes of PCa. These preliminary findings indicate that further follow-up experiments should be evaluated, including the molecular evaluation of CAMLs as reservoirs of biological information regarding tumor pathogenesis, and if CAML cells have any clinically useful relationships to treatment strategies.

## Figures and Tables

**Figure 1 cancers-15-03725-f001:**
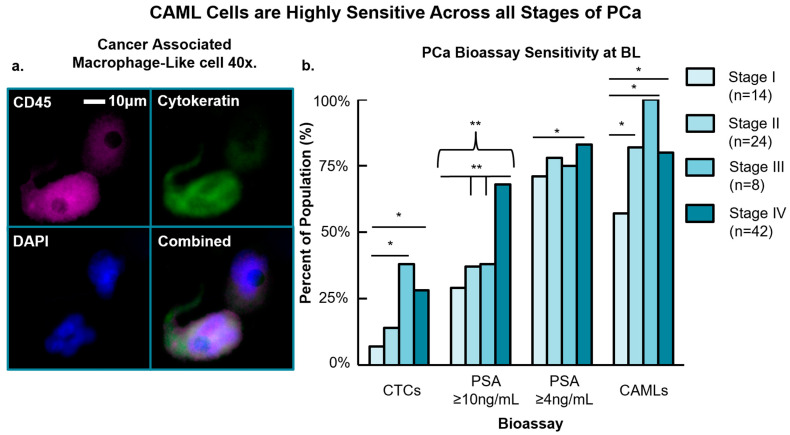
(**a**) Cancer-associated macrophage-like cells under 40× magnification. CAMLs are identified based on their size, polyploid nuclei, CD45 and DAPI positivity, and diffuse expression of cytokeratin. (**b**) Distribution of different PCa blood biomarkers and their sensitivity based on pathological stage. Overall, CAMLs were found to be the most sensitive assay, followed by PSA ≥ 4 ng/mL. Black brackets compare non-metastatic versus metastatic patients. Individual black lines compare patients of different pathological stage. Single-factor ANOVA was used to determine statistical differences for each bioassay among each pathological stage. Single asterisk (*) indicates statistical trending (*p* ≤ 0.15 and >0.05). Double asterisks (**) indicate a statistically significant difference in assay presence between groups (*p* ≤ 0.05).

**Figure 2 cancers-15-03725-f002:**
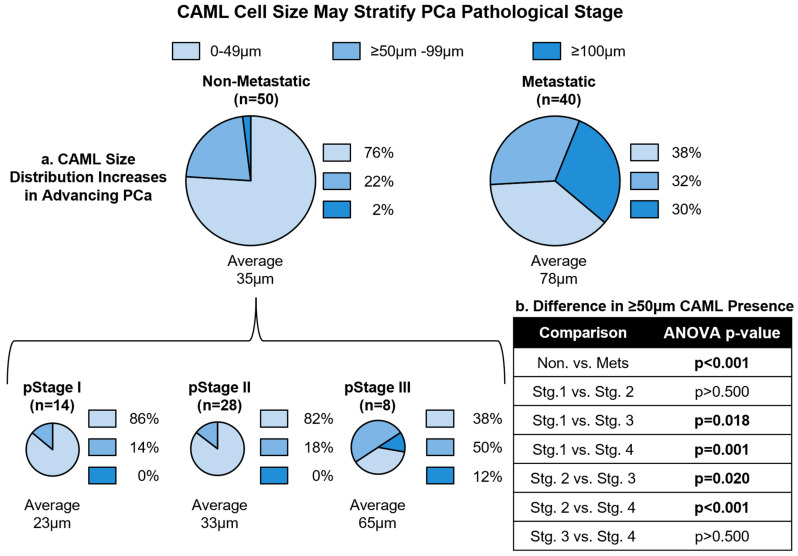
(**a**) CAML size distribution increases in advancing PCa. Non-metastatic disease is further broken up into pathological stages I, II, and III to demonstrate CAML size increases with advancing disease. (**b**) Statistical analyses. Single-factor ANOVA was used to compare the difference in ≥50 μm CAML frequency between non-metastatic and metastatic patients, as well as among each pathological stage.

**Figure 3 cancers-15-03725-f003:**
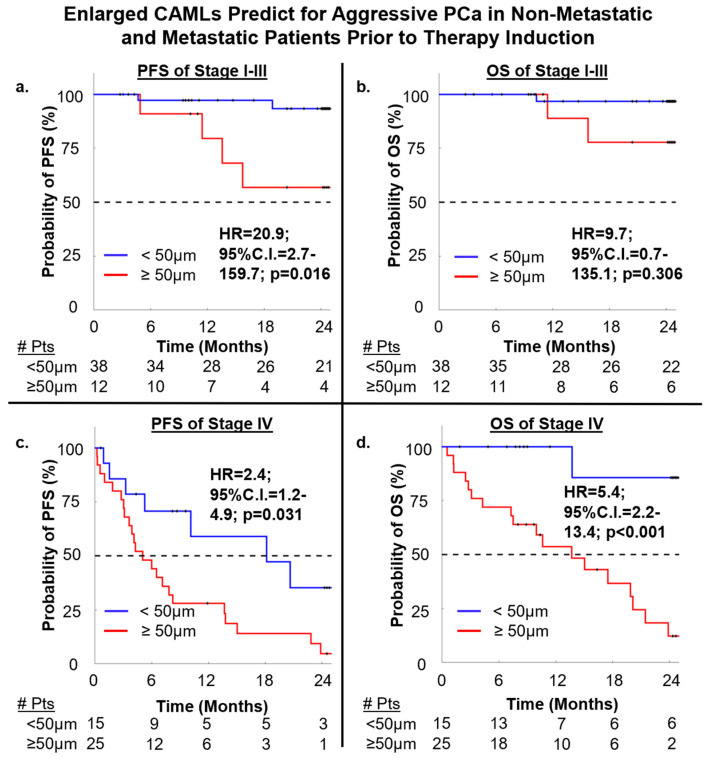
Log-rank analysis of PFS and OS based on CAML size at BL. (**a**) PFS stage I–III pts. (**b**) OS stage I–III pts. (**c**) PFS stage IV pts. (**d**) OS Stage IV pts.

**Figure 4 cancers-15-03725-f004:**
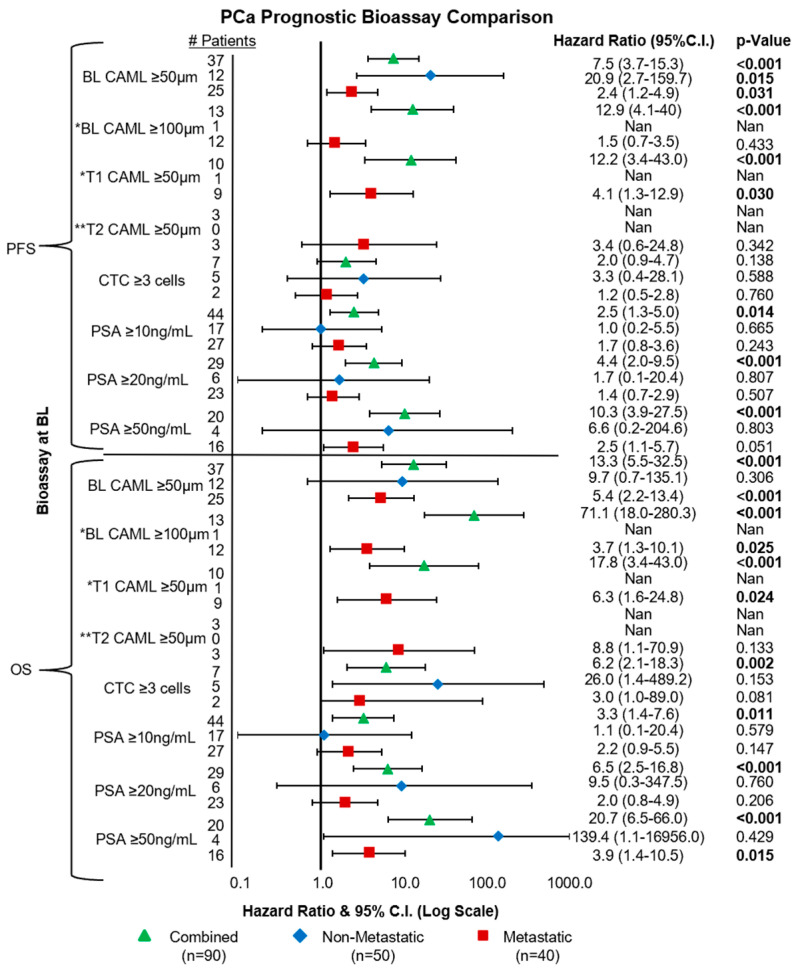
Forest plot comparing CAMLs ≥50 μm, CAMLs ≥100 μm, PSA ≥10 ng/mL, PSA ≥20 ng/mL, and PSA ≥50 ng/mL for predicting PFS and OS at BL. Log-rank survival analysis for ≥100 μm CAMLs could not be run for non-metastatic patients due to lack of samples. * Log-rank analysis could not be run in the non-metastatic setting due to too few patients reaching endpoint. ** Only metastatic patient follow-up data were available.

**Table 1 cancers-15-03725-t001:** Patient demographics.

Demographic	Non-Metastatic (n = 50)	Metastatic (n = 42)	Combined (n = 92)
Age (Years): Median (Range)	66 [50–81]	73 [48–89]	69 [48–89]
Race			
Caucasian	43 (86%)	37 (89%)	80 (87%)
African American	1 (2%)	2 (5%)	3 (3%)
Hispanic	1 (2%)	1 (2%)	2 (2%)
Asian	1 (2%)	1 (2%)	2 (2%)
Unknown	4 (8%)	1 (2%)	5 (6%)
Pathological Stage			
I	14 (28%)	0 (0%)	14 (15%)
II	28 (56%)	0 (0%)	28 (30%)
III	8 (16%)	0 (0%)	8 (9%)
IV	0 (0%)	42 (100%)	42 (46%)
Histology			
Adenocarcinoma	46 (92%)	39 (93%)	85 (92%)
Neuroendocrine	0 (0%)	1 (2%)	1 (1%)
Unknown	4 (8%)	2 (5%)	6 (7%)
Gleason Score			
6	2 (4%)	2 (5%)	4 (4%)
7	22 (44%)	14 (33%)	36 (39%)
8	14 (28%)	7 (17%)	21 (23%)
9	12 (24%)	15 (36%)	27 (30%)
10	0 (0%)	1 (2%)	1 (1%)
Unknown	0 (0%)	3 (7%)	3 (3%)
Received Prior Therapy			
Androgen Deprivation Therapy	14 (28%)	39 (93%)	53 (58%)
Chemotherapy	7 (14%)	30 (71%)	37 (41%)
Average BL PSA (ng/mL) (Median)	20.5 (7.1)	186.5 (30.3)	95.1 (9.4)
CAMLs Present (BL)			
Average/7.5 mL blood (Median)	3 (2)	6 (2)	5 (2)

## Data Availability

The data that support the findings of this study are available upon request from the corresponding authors, Daniel J. Gironda and Daniel L. Adams. The data are not publicly available due to restrictions such as information that could compromise the privacy of research participants.

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
