# Peer review of "Cancer Associated Macrophage-like Cells Are Prognostic for Highly Aggressive Prostate Cancer in Both the Non-Metastatic and Metastatic Settings"

_cancers, 2023, doi:10.3390/cancers15143725_

Round 1

Reviewer 1 Report

In this manuscript, Gironda and colleagues report the use of Cancer Associated Macrophage-Like cells as prognosis biomarker to discriminate indolent to aggressive prostate cancer (PCa). This discrimination is indeed a major issue in PCa Research.

To address this question, the authors have established a cohort of non-metastatic and metastatic PCa patients with blood samples for CTCs and CAMLs isolation. The presence of CAMLs has been correlated with different patient clinical features.

Discrimination of indolent from aggressive PCa is a majo issu fo PCa and CAMLs could serve as biomarkers in this intention. This study, as explained in the discussion, is quite preliminary with a lack of some controls. Especially, to compare PSA and CAML, an inclusion of BPH patients would have been done to demonstrate the absence of CAML with an increase in PSA.

In general the quality of figures is low (blurred). One title per figure is enough and should be included in the legend. For easier reading, a conclusion should be added at the end of each part of the result.

The definition of PCa stages should be added in the materials and methods

In figure 2 the authors have decided to only present the divided group of CAMLs sizes. Here again, for easier understanding, the curves of the size repartitions should be added.

Concerning the group of sizes, the authors should explain the choice of these groups.

The authors demonstrate the PFS ad OS correlated with CAMLs in samples before, during and after treatment. Is there any information of the treatment administrated to these patients ? Is there a link between the presence/size of CAMLs and treatment resistance/sensitivity ?

Reviewer 2 Report

Comments:

1) Introduction p2, line 55-56 - 'measuring rising PSA leverages PSA kinetic measurements'. This sentence needs revising

2) Figure 1a, please provide a negative or isotype control for cytokeratin, the staining is very weak and hard to see. 

2) Figure 1b, I find the asterisks for statistically trending, but not significant, misleading and would recommend removing it. 

3) Figure 1b,  why is PSA >10ng/ml not the more sensitive assay when it is statistically significant, compared to CAMLs and PSA >4ng/ml?

4) Page 6, lines 232-240 - what data is this referring to? 

5) Page 8, lines 279-285 - this data isn't included in Figure 3 as cited, its in Supplementary Figure 2 (a-b)

6) Page 13, lines 406-407 - what about the other 6 patients? Why have they been excluded?

7) Supplementary Figure 6. What were your cut offs for PFS and OS to determine whether CAMLs were more accurate at predicting survival than PSA tracking. It seems very arbitrary - patient 2 has an OS of 21.4 months, compared to patient 3 and patient 4, with an OS of 24.0 months. Furthermore, for patient 2, the CAML size = 48 um, which is less than 50 um and decreasing from the previous measurements. So how is this more accurate at predicting disease progression compared to PSA? It is not that much bigger than CAML size (41 um) at T2 for patient 3. Also, is this the average CAML size at each time point? Please explain in figure legend. 

8) Please provide n numbers and abbreviations in figure legends. 

9) Discussion page 13, line 427 - this was not significant and should be noted

Round 2

Reviewer 1 Report

The authors have well respond to the questions.

Reviewer 2 Report

The authors have made the requested changes